# Factors Associated with Mortality with Tuberculosis Diagnosis in Indigenous Populations in Peru 2015–2019

**DOI:** 10.3390/ijerph192215019

**Published:** 2022-11-15

**Authors:** Hoover León-Giraldo, Oriana Rivera-Lozada, Elvis Siprian Castro-Alzate, Rula Aylas-Salcedo, Robinson Pacheco-López, César Antonio Bonilla-Asalde

**Affiliations:** 1Escuela de Posgrado, Universidad Libre Cali, Cali 760031, Colombia; 2South American Center for Education and Research in Public Health, Universidad Norbert Wiener, Lima 15046, Peru; 3Escuela de Rehabilitación Humana, Universidad del Valle, Cali 760000, Colombia; 4Estrategia Sanitaria Nacional de Prevención y Control de la Tuberculosis, Ministerio de Salud, Lima 07021, Peru; 5Instituto de investigación en Interculturalidad, Universidad Privada San Juan Bautista, Lima 07006, Peru

**Keywords:** tuberculosis, mortality, indigenous, logistic regression

## Abstract

Objective: To identify factors associated with mortality in indigenous populations diagnosed with tuberculosis in Peru, 2015–2019. Methods: We conducted a nested case-control study in a retrospective cohort using the registry of indigenous peoples of the National Health Strategy for TB Prevention and Control of the Ministry of Health of Peru. A descriptive analysis was performed, and then bivariate and multivariate logistic regression was used to evaluate associations between the variables and the outcome (alive–deceased). The results are shown as OR with their respective 95% confidence intervals. Results: The mortality rate of the total indigenous population of Peru was 1.75 deaths per 100,000 indigenous people diagnosed with TB. The community of Kukama Kukamiria-Yagua reported 505 (28.48%) individuals, followed by the Shipibo-Konibo community with 385. The final logistic model showed that indigenous males (OR = 1.93; 95% CI: 1.001–3.7) with a history of HIV prior to TB (OR = 16.7; 95% CI: 4.7–58.7), and indigenous people in old age (OR = 2.95; 95% CI: 1.5–5.7) were factors associated with a greater chance of dying from TB. Conclusions: It is important to reorient health services among indigenous populations, especially those related to improving a timely diagnosis and early treatment of TB/HIV co-infection, to ensure comprehensive care for this population considering that they are vulnerable groups.

## 1. Introduction

Tuberculosis (TB) is an infectious disease produced by species of the Mycobacterium tuberculosis complex that can affect any tissue, but mainly the lungs [1]. Although it has a worldwide distribution, there is a greater effect mainly in population groups with social vulnerabilities and high risks of transmission. Among them, we can mention extreme poverty, overcrowding, populations deprived of their liberty, social outcasts, street people, displaced persons, and those with clinical conditions that affect the immune system such as HIV, immunosuppression, and diabetes, which, likewise, affect indigenous populations globally [2].

During 2020, the PAHO/WHO reported a total of 9.9 million people worldwide who fell ill with TB, with an estimated 1.5 million deaths due to this infection, of which 214,000 had HIV. In the Americas, 291,000 TB cases were reported in 2020. The estimated number of deaths in the region was 27,000, 29% of which (7900) corresponded to a TB/HIV co-infection [3].

In 2018, 29 countries in the Americas region reported the number of TB cases diagnosed in indigenous populations. Ten of them reported that they had no TB cases in indigenous populations, and the remaining nineteen reported 11,608 cases, representing 7.0% of the total number of incident TB cases reported by these countries. Brazil, Guatemala, and Mexico accounted for 75.8% of TB cases in indigenous people [4].

A systematic review by Tollefson et al. [5] reported that several countries in the Americas had high incidences of TB in some of their indigenous peoples. For example, in the Ache community of Peru, the incidence was up to 75 times higher than in the general population. Similarly, the number of cases in Amazonian communities in Brazil exceeded 1000 cases per 100,000 inhabitants.

Peru accounts for 14% of the TB cases in the region of the Americas; the Lima metropolitan area and Callao report 64% of the country’s TB cases, 79% of multidrug-resistant TB (MDR-TB) cases, and 70% of extensively drug-resistant TB (XDR-TB) cases [6]. Even though these figures show that Peru is no stranger to this problem, a significant decrease in the notification of TB cases and other public health events has been observed during 2020, despite the efforts of health personnel to continually provide interventions. The reversal of this situation will not be feasible as long as the COVID-19 pandemic is not controlled.

Several studies have shown evidence that the incidence of active TB and the prevalence of latent TB are substantially higher in indigenous groups than in non-indigenous populations [5,7]. Therefore, it was important to highlight the characteristics of this target population in this study.

It is estimated that there are approximately 476 million indigenous peoples around the world. Even though this figure is equivalent to only 6% of the world’s population, they represent about 19% of the extreme poor [8]. Indigenous populations have higher rates of extreme poverty, morbidity, and mortality than their non-indigenous neighbors across the spectrum of low- to high-income countries [9]. The scenario described above is conducive to the development of tuberculosis; therefore, efforts should be made to identify and treat it early. This can contribute to the decrease in its impact with interventions that progressively and sustainably reduce mortality and morbidity, as well as its social and economic repercussions in this population, who are often excluded from access to health services. In a study carried out in six indigenous populations in the U.S., Canada, and Greenland, it was found that BCG vaccination given to children, as well as the detection and treatment of LTBI, were associated with significant decreases in TB notification rates in these indigenous populations [10].

In Mexico, a study in indigenous populations reported that municipalities with a high proportion of indigenous people had the highest notification rates of all new TB cases [11]. In Brazil, a study reported the results of a cross-sectional survey aimed at detecting active TB in an indigenous group, the Suruí, with a very high annual incidence of notifications. According to the National Indigenous Health Service, the Suruí are among the top five indigenous groups in the country in terms of TB incidence [12]. In Colombia, an observational study was conducted in indigenous peoples of the Colombian Pacific in the department of Choco, where it was found that the incidence rate among the indigenous peoples in of this region, who represent 10% of the population, was 192.1 per 100,000 inhabitants, which was above the overall rate in Colombia (25.3 per 100,000 inhabitants on average) [13].

Among the studies conducted in Peru, one was undertaken in Asháninka communities in 2011, in which it was found that, in terms of gender and age, the most affected members of the indigenous population were adult men. This is similar to the TB data in the general population. Moreover, in terms of occupation, it was reported that the majority of the community was engaged in independent agricultural work, which is explained by the traditional and subsistence lifestyle of the Ashaninka communities. It is also important to note that the majority of cases resided in communities where there were no health facilities, which added to the difficulty of administering and supervising TB treatment [14].

A similar study in the same year, 2011, but in Quechua communities with 211 records of patients affected by TB, showed that the percentage of Quechua people with TB in the sample was 93.57% (95% CI: 88.15–97.02). The proportion of TB cases in the Quechua population was higher than 90% overall and by year. Thus, in 2008 the proportion was 94.23% (95% CI: 84.05–98.79); in 2009 it was 95.35% (95% CI: 84.18–99.43); and in 2010 it was 93.57% (95% CI: 78.78–97.52). In the indigenous Quechua communities, 87.5% of the cases were pulmonary TB and 11.7% were extrapulmonary TB, which is within the permitted range for this type of TB (10–15%). The higher value could probably be explained by the presence of HIV/AIDS, but no cases of HIV/AIDS were identified, in spite of the fact that the screening coverage with ELISA for HIV and confirmation with IFA or Western blot was probably low [15].

However, in recent years, in the Latin American context in general and in Peru in particular, evidence has been generated on the magnitude of the problem. This has focused mainly on morbidity, which generates a favorable scenario for analyzing fatal outcomes and suggesting factors that could explain it. Therefore, the objective of this study was to describe mortality due to TB diagnosis in the indigenous population in Peru and its associated factors during the period 2015–2019.

## 2. Materials and Methods

Study design. We conducted a nested case-control study with an epidemiological design in a retrospective cohort. This was characterized by analyzing all the cases of the stable cohort followed over time, and for the controls, the analysis was conducted on a sample of individuals from the same cohort [16].

Cases were defined as the indigenous population diagnosed with sensitive TB who died during the study period. In principle, the controls of the deceased cases should be alive, since they constitute a sample of the source population that gave rise to the cases [17].

Study Population and Selection. The general population with a sensitive TB diagnosis in Peru were 156,406 individuals, of which 4759 individuals belonging to the indigenous populations of Peru were selected during the period 2015–2019. The selection criteria were indigenous people diagnosed with TB; data mining was also performed. The database was cleaned considering missing data and outliers, which were corrected with the original sources. The name of the health facilities registered were validated and we excluded those with incomplete information on discharge status. After filtering and verifying the selection criteria, 2986 records were excluded due to having incomplete information, especially regarding discharge status, and we finally obtained a database with 1773 indigenous peoples of the Peruvian territory, of which 101 cases died.

Selection of controls. Based on the above, the statistical power was calculated for the different scenarios with 1, 2, and 3 controls per case and an OR of 2 to be detected; for the ratio of one control per case and an OR of 2.0, a statistical power of 84% was obtained to estimate differences. As the number of controls increased, the statistical power decreased, as shown in Figure 1.

The selection of an equal number of cases and controls allowed us to minimize the variance of the estimated odds ratio [18], which gave us a total of 101 cases (deceased) and 104 controls (alive), with which the logistic regression analyses were developed (see Figure 2).

Sampling design of controls (alive). The selection of the controls was carried out through a systematic sampling of the 1672 individuals of the indigenous population that were alive during the study period. Then, the fact that the proportion of individuals per year and per gender was conserved according to the population distribution was validated. Systematic sampling is a type of probability sampling where the first element is randomly selected for the sample, and then subsequent elements are selected using fixed or systematic intervals until reaching the desired sample size, which corresponded to 104 controls in this study. the fact that the proportion of individuals per year and per sex was conserved was validated according to the distribution of the population.

Data Variables and Source of information. We used secondary data, which were obtained from the registry of indigenous peoples of the National Health Strategy for TB Prevention and Control of the Ministry of Health of Peru. All the information was collected for epidemiological surveillance purposes and was obtained by personnel in charge of the TB program following national guidelines. The variables included were gender, age, life cycle, ethnic affiliation, location, admission condition, discharge condition, comorbidities and risk factors, and treatment regimen. The dependent variable was qualitative and dichotomous, i.e., yes or no, and was obtained from the discharge status.

There are two types of nested studies: simple ones and those that use incidence density. In our case, we carried out a simple nested case-control study, in which the answer to the event was infrequent (the frequency of indigenous deaths from TB was low), and it had a sufficient initial measurement. First, it identified all the participants of the cohort that had a final answer in the follow-up period (deceased), and then it defined a randomized sample of those who did not have a final answer (alive) [19]. Therefore, we did not estimate either incidence density or matching.

Statistical analysis. Descriptive analysis was performed for each of the study variables, both sociodemographic and clinical, and they were shown as percentages. The normality of the quantitative variables was contrasted by means of the Kolmogorov–Smirnov test and, if the assumption was met, the mean and standard deviation was used; otherwise, the median and interquartile range were used. All results are shown in tables and figures below.

Regarding discharge status, we used the response variable “status” (alive or deceased), and bivariate and multiple logistic regression was performed to evaluate associations between the variables by means of the stepwise backward procedure. The results were shown as OR with their respective 95% confidence intervals. For the multiple model, the variables with a *p*-value < 0.25 were selected as proposed by Hosmer and Lemeshow [20]. No survival analysis was performed in this study.

Finally, we evaluated the goodness-of-fit of the model through the likelihood ratio, comparing the difference between the two models (saturated and adjusted) as well as the Hosmer and Lemeshow test, thus defining that the model fit the data very well and was adequate in making predictions. The explanatory and predictive capacity of the factors in the model with respect to TB mortality was evaluated by means of the area under the curve, AUC.

The level of statistical significance was established at *p* < 0.05, and the Microsoft^®^ Excel^®^ program and the statistical package Stata corp. version 17.0^®^ (College Station, TX, USA, Stata Corp LLC. License from the Universidad Libre) were used for all analyses.

## 3. Results

The mortality estimated in this study was 5.7 cases per 100 indigenous individuals diagnosed with sensitive TB. However, the mortality rate calculated with respect to the total indigenous population of Peru was 1.75 deaths per 100,000 indigenous peoples (Table 1). A similar percentage behavior was observed in all years except in 2018 where it decreased to 4% of the total indigenous diagnosed.

Table 2 shows the demographic and clinical characteristics of the indigenous population studied. We observed that the median age was 39 years. In addition, 50% were between 26 and 55 years old, which corresponded to the interquartile range; 61.3% were male (1087). According to the life cycle stage, the majority of the indigenous population were adults (53%), corresponding to the age range between 27 and 59 years; this was followed by 20.6% who were in the old age stage (≥60 years). Most of the indigenous people (90%) had pulmonary TB and the remaining 10% had extrapulmonary TB.

In terms of ethnicity by community, the Kukama Kukamiria-Yagua community reported 505 individuals (28.48%), followed by the Shipibo-Konibo community with 385 individuals (21.7%), and in third place, the Asháninka community with 298 individuals, equivalent to 16.8%. These three communities accounted for 67% of the indigenous populations.

Most of the records of the indigenous population were new when being admitted (91.99%), and only 4.4% were admitted for relapse. On the other hand, the most frequent status of discharge was “cured” with 54.1%, followed by complete treatment with 21.2%. It was also observed that the proportion of deaths was 5.7% (101). Among the habits of the indigenous population, there was a low proportion of alcohol consumption (5.4%), smoking (4.2%), and drug addiction (2.5%). The majority of the indigenous population (91.9%) received treatment for sensitive TB, and the rest received treatment for HIV-TB and extrapulmonary TB.

The bivariate logistic regression analysis (Table 3) included gender, age grouped by life cycle stage, location of TB, previous history of HIV and DM, HIV result, diagnosis of DM through a glycemia test, and the habits of the indigenous population, namely alcoholism, smoking, and drug addiction. It was observed that male indigenous people had 2.14 times the chance of dying from TB compared to indigenous females (*p* = 0.012). It was evidenced that for a one-year increase in age, the probability of dying from TB was 1.7% (*p* = 0.017); when analyzing by life cycle stage, it was found that, for the indigenous people, being in an older age group was a risk factor for dying from TB compared to being indigenous minors, see Figure 3.

The indigenous population with a history of HIV prior to TB had 12.6 times the chance of dying from TB compared to those who did not have a previous history of HIV (*p* = 0.00). Likewise, those who tested positive for HIV had 20 times the chance of dying from TB compared to those who tested negative for HIV (*p* = 0.000).

In the multiple model, we evaluated the collinearity between the variables, HIV history, and HIV result. Independent analyses were performed, and finally the variable “History of HIV” was left because it had less imprecision in its estimates and also because of the total indigenous population with a history of HIV, 96% tested positive in a HIV test.

The adjusted or final multiple logistic regression analysis showed that, in Peru, indigenous men, increasing age, and a previous history of HIV were factors associated with a higher chance of dying from TB. The final model better explained the chance of dying from TB, which indicated a good goodness-of-fit of the model, and the Hosmer–Lemeshow test showed that the model fit the data well (*p* = 0.812) (Table 4). This model correctly classified 68.97% of the indigenous people with a probability of dying from TB, with a sensitivity of 51.5% and a specificity of 85.6%.

## 4. Discussion

In this research, we described the sociodemographic and clinical characteristics of the indigenous population of Peru diagnosed with TB. Moreover, we determined TB mortality during the period 2015–2019 and its possible associated factors.

In this study, we observed that the estimated mortality was 5.7 cases (101/1773) per 100 indigenous people diagnosed with TB. The mortality rate with respect to the total indigenous population of Peru was 1.75 deaths per 100,000 indigenous people, taking into account the 2017 census [21]. This rate was lower than what was reported in a study also conducted in Peru, where it was found that the specific mortality rate for pulmonary TB, in Aymara populations, was 11.78 per 100,000 inhabitants [22]. Furthermore, in a study on TB in indigenous communities in Choco Colombia, a rate of 3.1 deaths per 100,000 inhabitants was reported [13], and it was even much lower than what was reported in studies in Chile and Brazil, which corresponded to 14.4 [23] and 13.2 [24] per 100,000 indigenous people, respectively. The rate standardization method was not used in this study.

Male was the predominant gender in this study with 61.3%, and it was 7.6% higher compared to the study by Malacarne et al. [25] in 2016, where they found that the percentage of infected indigenous men was 53.7%. This was 4.3% higher with respect to the study by Ríos et al. [24], who reported TB mortality in indigenous men (57%); the percentage of deceased men with TB was also higher within the group of deceased indigenous people, it being 75% (76/101), which represented a rate of 2.7 per 100,000 inhabitants.

When observing TB mortality throughout the study period, a very slight upward trend was observed; however, in 2016 mortality decreased by 50% compared to 2015 (6% vs. 3%, respectively). In Paraguay, as part of the results obtained in the training of indigenous promoters for active search, a decrease in mortality from 26.5 in 2008 to 21.1/100,000 inhabitants in 2009 was achieved [26].

Through the bivariate analysis we could estimate the association between gender and TB mortality, thus we confirmed the aforementioned, which showed a statistically significant association: males had 2.14 times the chance of dying from TB compared to females. This result was also consistent with what was found in a study conducted in Venezuela, where it was mentioned that mortality and morbidity rates increased with age, and among the elderly, men were the most affected [27]. 

The multivariate analysis confirmed the significant association with gender, where men had 1.94 times the probability of dying from TB compared to women, adjusting for the other variables. One of the relevant results was the evaluation of TB/HIV co-infection, where indigenous people with a previous history of HIV were 16.5 times more likely to die from tuberculosis than those without a history of HIV. According to the literature, the risk of death due to TB in a patient with HIV is 2 to 4 times higher compared to a patient with TB and without HIV [28]. This result was also similar to what was reported in a study conducted in Paraguay, where TB mortality in all forms was investigated for the first time in the country [29].

The calculated area under the curve (AUC) showed a prediction of mortality in the indigenous population with a TB diagnosis of 73.3%. This result was similar to what was reported by Alves et al., who also used a logistic model to demonstrate the ability to discriminate the risk of death from tuberculosis of 75.6%; the model proposed by the authors considered aspects such as vulnerability, extreme poverty, and illiteracy in a Brazilian city with a high incidence of this disease [30].

On the other hand, it is important to mention that even though there is literature recommending the use of a certain number of controls (2, 3, or 4) for each case in order to increase the statistical power of the study [31,32], it should be noted that for a total sample size, the maximum power of a study is obtained with groups of equal sizes and the power decreases almost linearly with increasing unbalanced rates [33].

Due to the fact that, in nested case-control studies, information on risk factors of interest and the main variables are collected at the beginning of follow-up, prospectively, and before disease development, there is less risk of the classic information biases found in retrospective case-control studies.

Initially, we proposed to perform an analysis of survival from the time of initiation of treatment to the time of death in this study; however, as a limitation, the existing database had approximately 63% underreporting in the variables related to the recording of dates. This finding was consistent with what has already been described in a study of the Quechua communities in Peru, where the quality of the information from the health facilities was not good, given that there was a loss of records and incomplete records, which did not ensure reliable information [15].

The results of this study provided evidence about the association between certain factors and dying from TB; however, a causality could not be inferred, given that there are other studies such as clinical trials that represent the best methodological design for analyzing causality in research, as well as meta-analyses.

The absence of a good information system affects the quality of the data, and this was reflected in some laboratory results where the percentage of tests not performed, such as culture, was above 30%. This was also evidenced in a study in the Ashaninka communities, where there was a loss of relevant information due to poorly filled out medical records and reports, and the absence of information was up to 20% [14]. This was also evidenced in a study of the Quechua communities, where there are no adequate diagnostic methods; a little more than 50% perform only BK, and if there is a need to conduct a culture test, the sample has to be sent to the hospitals. In addition, in the case of drug sensitivity tests, they have to be sent to the health region or to the National Institute of Health. Since these communities are remote and the health facilities do not have the budget to transport samples, these procedures are omitted, which confirms why laboratory data are scarce [15].

Another important variable that could not be evaluated in this study was schooling. Nájera et al. [34], in their study on demographic and socioeconomic factors associated with pulmonary mortality, explain that low schooling can favor the occurrence of TB deaths because it is a determinant that increases the social vulnerability of the population.

In this study, it was not possible to evaluate the timeliness of care in the medical service because many of the variables related to this aspect had incomplete information (approximately 63%). Hence, it is necessary to conduct prospective studies to ensure the collection of these variables.

Despite the fact that TB is a treatable and curable disease and that no death from TB is acceptable, the data from this research showed that some vulnerable populations, such as indigenous peoples, are more affected by the disease, given that they have more social vulnerabilities that can be expressed in the barriers to access health services. Similarly, the cosmovision and cultural approach to the disease is a topic that should be studied in depth from a qualitative or mixed approach in order to refine and better understand the phenomenology of this population towards TB, as well as the perception of risk and stigma that may affect prevention and control strategies.

## 5. Conclusions

Our study showed that indigenous men with previous HIV history and in old age were more likely to die from TB. These factors could be used to reorient health services in indigenous peoples, incorporating parameters related to the improvement of timely diagnosis and early treatment in order to ensure comprehensive care for this population. Thus, we need to take into account the fact that they are vulnerable groups that require institutional responses through a differential approach that incorporate their worldview on the health–disease process.

According to PAHO’s guidelines for the prevention and control of TB in indigenous peoples, comprehensive care is explained as follows: “In indigenous populations, it is essential for traditional doctors—who are respected authority figures that represent the traditional practices component of integrated care (physical, emotional, and spiritual)—to be closely linked with support for treatment and adherence so that patients complete their pharmacological therapy. This starts with a knowledge dialogue that leads to the construction of shared care routes where traditional and Western medicine are jointly reflected (General guideline 2)”. [35].

In addition, to maintain the strengthening of collaborative actions on TB/HIV co-infection, it is advisable to improve the registry used for following up TB cases and to adequately use directly observed therapy (DOT), as well as to strengthen the quality of data on important variables such as barriers to access and treatment of TB.

## Figures and Tables

**Figure 1 ijerph-19-15019-f001:**
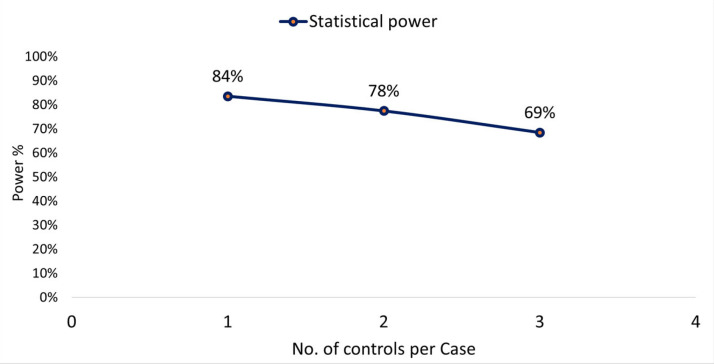
Relationship between number of controls per case and statistical power.

**Figure 2 ijerph-19-15019-f002:**
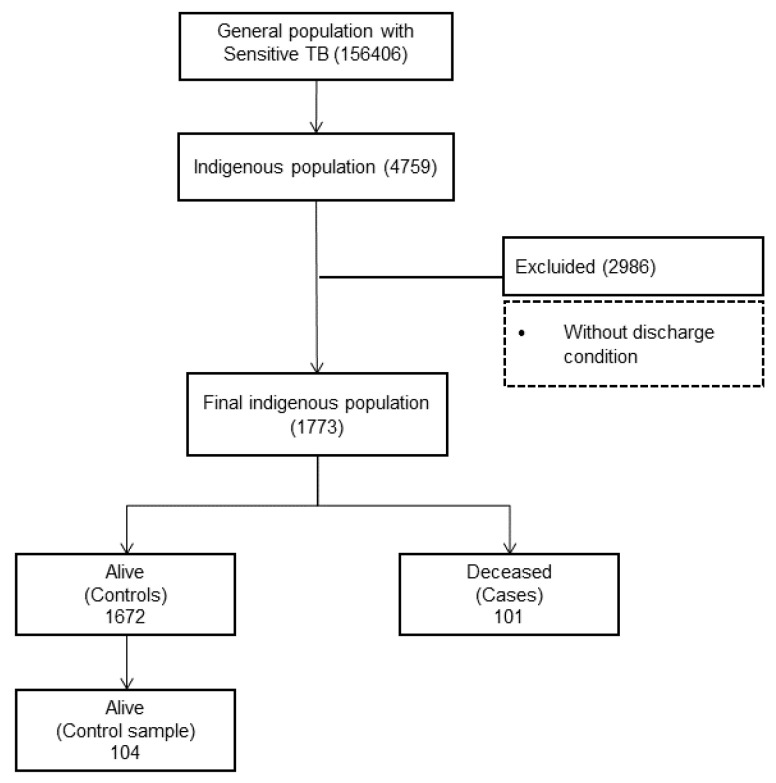
Flow chart of the study.

**Figure 3 ijerph-19-15019-f003:**
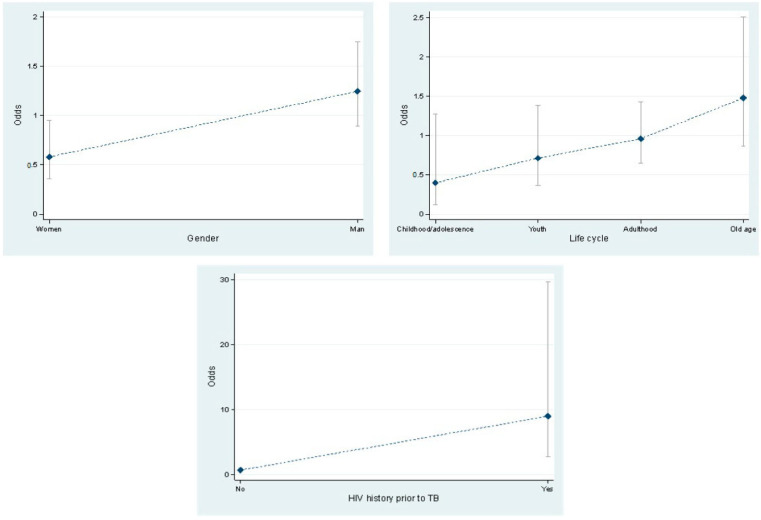
Association between gender, life cycle stage, HIV history, and TB mortality.

**Table 1 ijerph-19-15019-t001:** TB mortality by gender.

Mortality Rate	Indigenous Population (2017 Census)	Indigenous Deceased by TB	Total Indigenous d. w/TB	Mortality% by TB	Mortality Rate by TB (100,000 Inhab).
General	5,771,885	101	1773	5.7%	1.75
Male	2,801,412	76	1087	7.0%	2.71
Female	2,970,473	25	686	3.6%	0.84

**Table 2 ijerph-19-15019-t002:** Demographic characteristics of the indigenous population diagnosed with sensitive TB, 2015–2019.

Sociodemographic and Clinical Characteristics	Description	Summary Measure	
		*n*: 1773	%
Gender	Female	686	38.7
	Male	1.087	61.3
Age (years)	Median (IQR ^1^)	41 (26–55)	
Life cycle stage	Childhood/adolescence: (1 to 18 years)	140	8
	Youth: (>18 to 26 years)	327	18.4
	Adulthood: (>26 to 59 years)	940	53
	Old age: (>59 years old)	366	20.6
Ethnic affiliation	Kukama Kukamiria-Yagua	505	28.48
	Shipibo-Konibo	385	21.71
	Ashaninka	298	16.8
	Kichwa	117	6.6
	Kukama Kukamiria	109	6.15
	Ticuna	52	2.93
	Shawi	47	2.65
	Harakbut	37	2.09
	Kakataibo	36	2.03
	Bora	35	1.97
	AWAJÚN	33	1.9
	Matsigenka	32	1.8
	Achuar	18	1.02
	Yagua	18	1.02
	Wampis	16	0.9
	Mestizo	14	0.79
	Quechua	12	0.68
	Kandozi	4	0.23
	Sharanahua	3	0.17
	Madija	2	0.11
Location	Extrapulmonary	176	9.9
	Pulmonary	1597	90.1
Admission status	New	1631	91.99
	Relapsed	78	4.40
	Dropout recovered	61	3.44
	Failure	3	0.17
Discharge status	Cured	959	54.1
	Complete treatment	376	21.2
	Dropout	158	8.9
	Deceased	101	5.7
	Failure	5	0.28
	No answer	174	9.8
Comorbidities and risk factors			
HIV	Positive	126	7.47
Diabetes	Yes	170	11.1
Alcoholism	Yes	96	5.42
Smoking	Yes	74	4.2
Drug addiction	Yes	44	2.5
Treatment regimen ^2^	2HREZ/4(HR)3	1629	91.9
	2HREZ/10HR	43	2.4
	2HREZ/7HR	101	5.7

^1^ IQR: interquartile range; ^2^ treatment for sensitive TB: 4(HR)3; extrapulmonary TB: 10(HR)3; treatment for TB/HIV: 7(HR)3.

**Table 3 ijerph-19-15019-t003:** Association between mortality and sociodemographic and clinical factors of the indigenous community diagnosed with sensitive TB 2015–2019.

Characteristics		Deceased (101)	Alive (104)	OR (Crude)	95% CI		*p*-Value
Gender	Female *	25 (24.75)	43 (41.35)	1			0.012
	Male	76 (75.25)	61 (58.65)	2.14	1.8	3.9	
Life cycle stage	Childhood/ adolescence *	4 (3.96)	10 (9.6)	1			
	Youth	15 (14.8)	21 (20.2)	1.78	0.47	6.8	0.395
	Adulthood	48 (47.5)	50 (48.1)	2.4	0.7	8.2	0.16
	Old age	34 (33.7)	23 (22.1)	3.7	1.03	13.2	0.044
Location of TB	Extrapulmonary *	16 (15.84)	11 (10.58)	1			
	Pulmonary	85 (84.16)	93 (89.42)	0.63	0.28	1.43	0.27
HIV history prior to TB	No *	72 (72.73)	101 (97.12)	1			
	Yes	27 (27.27)	3 (2.88)	12.6	3.7	43.2	0.000
DM history prior to TB	No *	90 (94.74)	96 (96)	1			
	Yes	5 (5.26)	4 (4)	1.33	0.35	5.1	0.67
Diagnosis of DM with glycemia test	Negative *	71 (88.75)	81 (87.10)	1			0.74
	Positive	9 (11.25)	12 (12.90)	0.86	0.34	2.1	
Alcoholism	No *	91 (90.1)	98 (94.23)	1			0.28
	Yes	10 (9.9)	6 (5.77)	1.8	0.63	5.1	
Smoking	No *	97 (96.04)	101 (97.12)	1			
	Yes	4 (3.96)	3 (2.88)	1.4	0.3	6.4	0.67
Drug addiction	No *	100 (99.01)	103 (99.04)	1			0.98
	Yes	1 (0.99)	1 (0.96)	1.03	0.06	16.7	

* Reference category.

**Table 4 ijerph-19-15019-t004:** Multiple logistic model and associated factors.

Characteristics		Deceased (101)	Alive (104)	OR (Crude)	95% CI		OR (Adjusted)	95% CI		*p*-Value
Gender	Female *	25 (24.75)	43 (41.35)	1			1			0.047
	Male	76 (75.25)	61 (58.65)	2.14	1.8	3.9	1.94	1.001	3.7	
Life cycle stage	Childhood/ Adolescence *	4 (3.96)	10 (9.6)	1						0.001
	Old age	34 (33.7)	23 (22.1)	3.7	1.03	13.2	2.95	1.5	5.7	
VIH history prior to TB	No *	72 (72.73)	101 (97.12)	1			1			0.000
	Yes	27 (27.27)	3 (2.88)	12.6	3.7	43.2	16.7	4.7	58.7	
Comparison of logistic regression models				Likelihood				*p*-value		
Saturated Model (5 covariates)				−119.867				0.9846		
Adjusted Model (3 covariates)				−120.054						
Hosmer–Lemeshow’s goodness-of-fit test				X^2^: 0.21				0.9949		

* Reference category.

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
