# Peer review of "Factors Associated with Mortality with Tuberculosis Diagnosis in Indigenous Populations in Peru 2015–2019"

_ijerph, 2022, doi:10.3390/ijerph192215019_

Round 1

Reviewer 1 Report

León et al. aimed to identify factors associated with mortality with tuberculosis diagnosis in the in-17 digenous population in Peru 2015-2019. It is an interesting article but there are some shortcompings that need attention. I would like to make some suggestions and contributions to the present manuscript.

METHODS

In line 121-122, mention "After filtering and verifying the selection criteria...". Explicitly specify the selection criteria for the study.

In sources of information, provide more details on how the data were obtained. Provide more details of the study procedure.

Mention the variables used for matching.

Add variables section. It is important to explicitly mention the operational definition of the dependent variable and the independent variables. For example, it is important to know how the measurement of clinical variables (location, comorbidities, risk factors, therapeutic regimen) was obtained from the data collection source. Additionally, it is necessary to describe the therapeutic scheme; for unfamiliar readers it would be helpful to know what 2HREZ/4(HR)3, 2HREZ/10HR, 2HREZ/7HR, 2HREZ/7HR means.

In analysis plan, mention if collinearity was assessed in the variables of interest. I have a question about the analysis, if the database was formatted for survival analysis (using stset) and if it was then stated that it is a nested case-control study with paired variables (using sttocc command). In this nested control case was incidence density sampling used? If so, remember that the OR would be interpreted as an approximation of the Hazard Ratio (HR), which represents the instantaneous risk associated with an exposure.

Add section on ethical aspects

RESULTS

Table 2 shows age categorized. Show the ranges used to categorize the age groups in methods.

In results paragraphs of Table 3 and 4, it is more important to include OR and 95%CI values for significant variables, replacing p-values.

DISCUSSION

Revise the first discussion paragraph (lines 232-235). It should be omitted.

Lines 236-238 should be revised. It does not seem to contribute to the discussion.

In the third paragraph, where TB mortality in indigenous population is discussed. In limitations section, it should be added that the given rate standardization method that allows comparison of frequency measures of an event in different populations while holding potentially confounding characteristics, such as age, constant, was not used.

The crude OR of the association between gender and death is discussed on lines 259-264. The adjusted OR should be discussed.

On lines 265-271, the association between HIV and death is discussed. Although the association was significant, this finding should be interpreted with greater caution given that the proportion of COVID-HIV patients who died is small (n=3). Therefore, the confidence intervals are wide and lack precision.

It is necessary to explicitly mention which were the methodological biases of the study in the limitations paragraph: selection bias, information bias, etc. Can causality be inferred from the association found? Additionally, discuss the generalisability (external validity) of the study results. In addition, the strengths of your study should be included. What are the advantages of your study compared to a case-control study? For example, minimize selection bias and recall bias.

A public health relevance of findings paragraph would enrich your discussion. I suggest you include.

Author Response

Evaluador 1.

Métodos

  1. On line 121-122, mention "After filtering and checking the selection criteria...". Explicitly specify the selection criteria for the study.

  • The selection criteria we used was the following: Indigenous people diagnosed with TB.
  • Data mining was performed. The database was cleaned, taking into account missing data, outliers, which were corrected with the original sources. The name of the health facilities registered were validated and we excluded those with incomplete information on discharge status.

  1. Under information sources, provide more details on how the data was obtained. Please provide more details of the study procedure.
  • We used secondary data and were obtained from the registry of indigenous peoples of the National Health Strategy for TB Prevention and Control of the Ministry of Health of Peru. All the information was collected for epidemiological surveillance purposes and was obtained by personnel in charge of the TB program, following national guidelines.

  1. List the variables used for matching.

  • There are two types of nested case-control studies. First, simple ones, in which the answer of the event is infrequent. Second, when an initial measurement of the exposure is sufficient, as in this study, in which the frequency of indigenous people deceased due to TB was low; hence, no variables were used.

  • Add variables section. It is important to explicitly mention the operational definition of the dependent variable and the independent variables. For example, it is important to know how the measurement of clinical variables (location, comorbidities, risk factors, therapeutic regimen) was obtained from the data collection source

  • The dependent variable is qualitative and dichotomous, Yes or No, and were obtained from the discharge status.

  • Additionally, it is necessary to describe the therapeutic scheme; For unfamiliar readers, it would be helpful to know what 2HREZ/4(HR)3, 2HREZ/10HR, 2HREZ/7HR, 2HREZ/7HR mean.

  • Treatment for sensitve TB: 4(HR)3; Treatment for TB/HIV: 7(HR)3; Extrapulmonary TB: 10(HR)3

  1. In the analysis plan, mention if collinearity was evaluated in the variables of interest. I have a question about the analysis, if the database was formatted for survival analysis (using stset) and then indicated that it is a nested case-control study with paired variables (using the sttocc command). In this nested control case, was incidence density sampling used? If so, remember that the OR would be interpreted as an approximation of the Hazard Ratio (HR), which represents the instantaneous risk associated with an exposure
  • Regarding collinearity assessment, it was mentioned in lines 224 to 226: “we evaluated collinearity between the variables, HIV history, and HIV testing results; then, we performed separate analyses, to finally keep "HIV history”.
  • No survival analysis was performed in this study.
  • As it was mentioned above, there exist two types of nested studies: simple ones and those who use incidence density. In our case, we carried out a simple nested case-control study, in which the answer of the event is infrequent and a sufficient initial measurement. First, it identifies all the participants of the cohort that have the final answer of the follow-up period (deceased), and then, it defines a randomized sample of those who did not have it (alive) (1). Therefore, we did not estimate incidence density nor matching.

  1. Add section on ethical aspects

It was mentioned in lines 345 to 347.

Resultados

  1. Table 2 shows the categorized age. Show the ranges used to categorize the age groups in the methods.

-Infancy/Adolescence: from 1 to 18 years old

-Youth: >18 to 26 years old

-Adulthood:>26 to 59 years old

- Old age: >59 years old

  1. In the results paragraphs of Tables 3 and 4, it is most important to include OR and 95% CI values for significant variables, replacing p-values.

Tables 3 and 4 show the crude and adjusted OR values, together with the 95% IC and p-values.

Discusión

  1. Review the first discussion paragraph (lines 232-235). must be omitted. MODIFIED IN THE ARTICLE.

MODIFIED IN THE ARTICLE.

  1. lines 236-238 should be reviewed. Doesn't seem to contribute to the discussion.

 CHECK IF WE KEEP IT OR DELETE IT.

  1. In the third paragraph, where TB mortality in the indigenous population is discussed. In the limitations section, it should be added that the rate standardization method was not used since it allows comparison of frequency measures of an event in different populations while keeping potentially confounding characteristics constant, such as age

MODIFIED IN THE ARTICLE.

  1. The crude OR of the association between gender and death is discussed in lines 259-264. The adjusted OR should be discussed.

MODIFIED IN THE ARTICLE.

  1. In lines 265-271, the association between HIV and death is discussed. Although the association was significant, this finding should be interpreted with greater caution given that the proportion of patients with COVID-HIV who died is small (n=3). Therefore, the confidence intervals are wide and lack precision.
  • In this study, we did not assess the association between COVID-HIV. In addition, the proportion of patients with previous history of HIV that died from TB were 27, the 3 mentioned are the alive ones.

  1. It is necessary to explicitly mention the methodological biases of the study in the limitations section: selection bias, information bias, etc. Can causality be inferred from the association found? Also, discuss the generalizability (external validity) of the study results. In addition, the strengths of your study should be included. What are the advantages of your study compared to a case-control study? For example, minimize selection bias and recall bias.
  • Due to the fact that in nested case-control studies information on risk factors of interest and the main variables have been collected at the beginning of follow-up, prospectively and before disease development, there is less risk of the classic information biases of retrospective case-control studies.
  • The results of this study provide evidence about the association between certain factors and dying from TB; however, causality cannot be inferred, given that there are other studies such as clinical trials that represent the best methodological design for analyzing causality in research, as well as meta-analyses.

  1. A paragraph on the public health relevance of the findings would enrich your discussion.

- Despite the fact that TB is a treatable and curable disease and that no death from TB is acceptable, the data from this research show that some vulnerable populations, such as Indigenous peoples, are more affected by the disease, given that they have more social vulnerabilities that can be expressed in the barriers to access health services. Similarly, the cosmovision and cultural approach to the disease is a topic that should be studied in depth from a qualitative or mixed approach in order to refine and better understand the phenomenology of this population towards TB, as well as the perception of risk and stigma that may affect prevention and control strategies.

Reviewer 2 Report

This is a very important study, and will inform public health policy in Peru.  It is important to have determined that risks for poor tuberculosis outcome are the same in the indigenous population of Peru as in populations elsewhere: HIV and old age.  I would expect malnutrition to also influence TB mortality but perhaps that is not a concern in these peoples. 

There are a number of areas which were unclear to this reviewer, mainly due to syntax and sometimes lack of full explanation.  A good editor should be able to help improve the clarity of the report. 

Examples of areas to clarify:
Introduction: 8th paragraph, first sentence. This is a very long sentence, and there is an unclear reference to childhood BCG vaccination. Last sentence in that paragraph: As you point out the % of the population represented by the indigenous people, it would be helpful to compare the TB rate to that of the overall population.  Is it higher?

Paragraph 9: This is unclear to me. 

Paragraph 10: Are 93% of Quechua infected with tuberculosis?

Materials and Methods: Paragraph 6: Please explain what systematic sampling is. Did you select the controls at random from the TB infected individuals? Or did you match on any factors?

Results Line 208: "indigenous people who are in an age group". Which age group?

Lines 210-213: Please explain the difference between "history of HIV" and "Tested positive for HIV". 

Discussion: Lines 232-235: This paragraph reads like instructions from someone and could be omitted. 

Lines 248-252: I am not sure what the percents are referring to here.  I assume for example what is the 75% in the last line referring to? 

I think this is a good study, demonstrating important information.  Some rewriting will help make the message clear. Thank you for your good work. 

Author Response

Evaluador 2.

  1. introduction: 8th paragraph, first sentence. This is a very long sentence, and there is an unclear reference to childhood BCG vaccination.CHECK IF WE KEEP OR ELIMINATE THIS SENTENCE.

  1. Last sentence in that paragraph: Since you point out the % of the population represented by indigenous peoples, it would be useful to compare the TB rate with that of the general population. Is higher?

MODIFIED IN THE ARTICLE.

  1. Paragraph 10: Are 93% of Quechuas infected with tuberculosis?

IT HAS BEEN EDITED.

  1. Paragraph 10: Are 93% of Quechuas infected with tuberculosis?

YES

  1. . Materials and Methods: Paragraph 6: Explain what systematic sampling is. Did you randomly select controls from TB-infected individuals? Or did you agree on some factor?

- Systematic sampling is a type of probability sampling where a random selection is made of the first element for the sample, and then subsequent elements are selected using fixed or systematic intervals until the desired sample size is reached, which in this study corresponded to 104 controls. It was validated that the proportion of individuals per year and per sex was conserved according to the distribution of the population.

 Systematic sampling is a type of probability sampling where the first element is randomly selected for the sample, and then subsequent elements are selected using fixed or systematic intervals until reaching the desired sample size, which corresponds to 104 controls in this study. It was validated that the proportion of individuals per year and per sex was conserved according to the distribution of the population.

  1. Results Line 208: “indigenous who are in an age group”. What age group?

 MODIFIED IN THE ARTICLE.

  1. Lines 210-213: Explain the difference between "history of HIV" and "positive result for HIV".

Of the total of Indigenous people with previous history of HIV, 96% tested positive for the HIV test.

  1. Discussion: Lines 232-235: This paragraph reads like instructions from someone and could be left out

MODIFIED IN THE PAPER.

  1. lines 248-252: I'm not sure what the percentages refer to here. I guess, for example, what is the 75% in the last line referring to?

It means that of the total of Indigenous people deceased due to TB, 75% (76/101) are male.

  1. I think this is a good study, showing important information. A bit of rewriting will help clarify the message. Thanks for your good work.

BIBLIOGRAPHY

  1. Alternative statistical methods and their application to research in Intensive Care. Intensive medicine. 2018 Nov 1;42(8):490-9.

Round 2

Reviewer 1 Report

I congratulate the authors for improving the manuscript with the suggestions/comments submitted.

Author Response

The observations were incorporated into the paper:

The treatment schemes were included and specified and it was checked that all the explanations were in the paper.
